# On Transmitted Complexity Based on Modified Compound States

**DOI:** 10.3390/e25030455

**Published:** 2023-03-05

**Authors:** Noboru Watanabe

**Affiliations:** Department of Information Sciences, Tokyo University of Science, Noda City 278-8510, Chiba, Japan; watanabe@is.noda.tus.ac.jp

**Keywords:** quantum dynamical mutual entropy, quantum entanglement, quantum compound system

## Abstract

Based on the classical dynamical entropy, the channel coding theorem is investigated. Attempts to extend the dynamical entropy to quantum systems have been made by several researchers In 1999, Kossakowski, Ohya and I introduced the quantum dynamical entropy (KOW entropy) for completely positive maps containing an automorphism describing the time evolution. Its formulation used transition expectations and lifting in the sense of Accardi and Ohya and was studied as a measure of the complexity of quantum mechanical systems. This KOW entropy allowed the extension of generalized AF (Alicki and Fannes) entropy and generalized AOW (Accardi, Ohya and Watanabe) entropy. In addition, the S-Mixing entropy and S-mixing mutual-entropy were formulated by Ohya in 1985. Compound states are an important tool for formulating mutual entropy, and the complexity was constructed by the generalized AOW entropy. In this paper, the complexity associated with the entangled compound states in the C* dynamical system based on the generalized AOW entropy based on the KOW entropy is investigated to lay the foundation for the proof of the theorem of channel coding for quantum systems. We show that the fundamental inequalities of the mutual entropy are satisfied when the initial state is transmitted over the channel changes with time.

## 1. Introduction

Shannon [1] paid a lot of attention to the mathematical treatment of communication systems, including the entropy of the system and the mutual entropy determined by the relative entropy of the joint probability distribution between input and output estimated by the channel and the direct product distribution between input and output. He introduced information measures and formulated information theory. Based on information theory, various researchers have studied the efficiency of information transmission through communication channels from input systems to output systems. In order to rigorously investigate the information transfer efficiency in optical communication, it is necessary to formulate a quantum information theory that can describe quantum effects.

The study of extending entropy to quantum systems was started by von Neumann [2] in 1932, and quantum relative entropy was introduced by Umegaki [3] and extended to more general quantum systems by Araki [4,5], Uhlmann [6] and Donald [7]. One of the key problems in quantum information theory is to investigate how accurately information is transmitted when an optical signal passes through an optical channel. To achieve this, we need to extend the mutual entropy determined in classical systems to quantum systems.

The mutual entropy of classical systems is defined using the joint probability distribution between the input and output systems, but it has been shown that joint probability distributions generally do not exist for quantum systems [8]. In order to determine the quantum mutual entropy in the quantum communication process, Ohya introduced a compound state (Ohya compound state [9,10]) that expressed the correlation between the initial state and the output state for the quantum channel. Using the quantum relative entropy, Ohya [9,10] formulated the quantum mutual entropy [9,10,11,12,13,14,15,16,17,18,19]. This quantum mutual entropy has been proved to satisfy the Shannon type inequality. This quantum mutual entropy was defined by Ohya in a C* dynamical system as the C* mixing entropy [13,20], which was extended and completely proven as the C* mixing Rényi entropy in [21]. These entropy properties have been studied in the literature [12,14,21,22,23]. Various studies have been conducted on the channel capacity [16,17,24] of quantum systems based on quantum mutual entropy. Entangled states [25,26] are one of the important themes in studying quantum information theory, and one of these notable results in discussing entangled states is Jamiolkowski’s isomorphism [27]. Important discussions have been made on the relationship between classical channels, including Gaussian channels, and quantum communication theory [28].

The study of dynamical entropy of quantum systems was started by Emch [29], Connes and Størmer [30], and various research works have been developed [29,30,31,32,33,34,35,36,37,38].

Based on Accardi’s transition expectation and lifting, the KOW entropy [36] was defined, and the AOW (Accardi, Ohya and Watanabe) and AF (Alicki and Fannes) entropies were generalized by the formulation of the KOW (Kossakowski, Ohya and Watanabe) entropy. Transmitted entropy [19] is defined based on the compound state. The generalized AOW entropy defines the transmitted complexity [39] with separable compound states. We introduced the hybrid compound state in [19]. This compound state did not use the possible decomposition of all states (see Section 3). Therefore, we introduced a new compound state called a modified compound state in [19]. We discussed the transmitted complexity for the modified compound states in dynamical systems described by the Hilbert spaces [40].

In this paper, in C* dynamical systems, we define the transmitted complexity by the modified compound state. We show the inequalities of the transmitted complexity for the C*-system.

## 2. Quantum Channels and Entropy and Mutual Entropy for General Quantum Systems

Let $A_{1}$$\left(\mathrm{respectively},\;A_{2}\right)$ be a C${}^{*}$-algebra and $S(A_{1})$$\left(\mathrm{respectively},\;S(A_{2})\right)$ be the set of all normal states on $A_{1}$ (respectively, $A_{2}$). We describe the input (respectively, output) quantum system by ($A_{1}$, $S(A_{1})$) (respectively, $(A_{2}$, $S(A_{2}))$. Λ is a linear mapping from $A_{2}$ to $A_{1}$ with $\Lambda \left(I_{2}\right)=I_{1}$, where $I_{k}$ is the identity operator $\left(i.e.,AI_{k}=I_{k}A=A,\left(\forall A\in A_{k}\right)\right)$ in $A_{k}$ (k=1,2). The dual map $\Lambda^{*}$ of Λ is a linear quantum channel from $S(A_{1})$ to $S(A_{1})$ given by $\Lambda^{*}\left(\varphi \right)\left(B\right)=\varphi \left(\Lambda (B)\right)$ for any $\varphi \in S(A_{1})$ and any $B\in A_{2}$. If Λ holds
$$\sum_{i,j=1}^{n}A_{i}^{*}\Lambda \left(B_{i}^{*}B_{j}\right)A_{j}\geq 0$$
for all n∈N, all $B_{j}\in A_{2}$ and all $A_{j}\in A_{1}$ is called a completely positive (C.P.) channel [12,14,17,18,41,42].

We here briefly explain the S-mixing entropy of general quantum systems [10,13,14,18,20,23,42].

Let A be a C*-algebra, we denote S(A) the set of all normal states on A. We express a weak* compact convex subset of S(A) by S. For any state φ∈S there is a maximal measure μ pseudosupported on exS such that
(1)$$\varphi =\int_{\left(exS\right)}\omega d\mu ,$$
where exS is the set of all extreme points of S. Let μ be a measure satisfying the above decomposition. It is not unique unless S is a Choquet simplex. Let $M_{\varphi }\left(S\right)$ be the set of all such measures and $D_{\varphi }\left(S\right)$ be a subset of $M_{\varphi }\left(S\right)$ such that
(2)$$\begin{array}{ccc} D_{\varphi }\left(S\right) & = & \left\{M_{\varphi }\left(S\right);\;\exists \mu_{k}\subset R^{+}\;\mathrm{and}\;\left\{\varphi_{k}\right\}\subset exS\right.\\ & & \left.s.t.\;\sum_{k}\mu_{k}=1,\;\mu =\sum_{k}\mu_{k}\delta \left(\varphi_{k}\right)\right\}, \end{array}$$
where δ(φ) is the Dirac measure centered on the initial state φ. Let *H* be the function
(3)$$H\left(\mu \right)=-\sum_{k}\mu_{k}\log\mu_{k}$$
for the measure $\mu \in D_{\varphi }\left(S\right)$. The S-mixing entropy of a state φ∈S(A) with respect to S is defined as
(4)$$S^{S}\left(\varphi \right)=\left\{\begin{array}{c} \inf\left\{H\left(\mu \right);\;\mu \in D_{\varphi }\left(S\right)\right\}\\ +\infty \;\;\;\mathrm{if}\;D_{\varphi }\left(S\right)=Ø. \end{array}\right.$$

It describes the amount of information of the state φ measured from the subsystem *S*. For example, if *S* is given by S(A), the set of all states on A, then describe $S^{S(A)}\left(\varphi \right)$ by S(φ), which means the natural extension of the von Neumann entropy [2]. If A is given by $B\left(H\right)$ and any $\varphi \in \;S\left(A\right)$, given by $\varphi \left(\cdot \right)=tr\varrho \left(\cdot \right)$ with a density operator ρ, then
S(φ)=−trρlogρ.

Here, I briefly review the mutual entropy of the $C^{*}$-system defined by Ohya [13]. For any $\varphi \in S\subset S(A_{1})$ and quantum channel $\Lambda^{*}:S\left(A_{1}\right)\to S\left(A_{2}\right)$, the compound states are defined as
$$\begin{array}{ccc} \Phi_{0} & = & \varphi ⊗\Lambda^{*}\varphi ,\\ \Phi_{\mu }^{S} & = & \int_{\left(exS\right)}\omega ⊗\Lambda^{*}\omega d\mu . \end{array}$$

The compound state $\Phi_{\mu }^{S}$ generalizes the joint probabilities of the classical system and shows the correlation between the initial state φ and the final state $\Lambda^{*}\varphi$. The mutual entropy with respect to *S* is defined as
$$I^{S}\left(\varphi ;\Lambda^{*}\right)=\lim_{\varepsilon \to 0}\sup\left\{I_{\mu }^{S}\left(\varphi ;\Lambda^{*}\right);\mu \in F_{\varphi }^{\varepsilon }\left(S\right)\right\},$$
where $I_{\mu }^{S}\left(\varphi ;\Lambda^{*}\right)$ is the mutual entropy with respect to *S* and μ described by
$$I_{\mu }^{S}\left(\varphi ;\Lambda \right)=S\left(\Phi_{\mu }^{S},\Phi_{0}\right).$$

$S\left(\Phi_{\mu }^{S},\Phi_{0}\right)$ represents the quantum relative entropy according to Araki [4,5] or Uhlmann [6].
$$F_{\varphi }^{\varepsilon }\left(S\right)=\left\{\begin{array}{c} \left\{\mu \in D_{\varphi }\left(S\right);S^{S}\left(\varphi \right)\leq H\left(\mu \right)\leq S^{S}\left(\varphi \right)+\varepsilon <+\infty \right\}\\ M_{\varphi }\left(S\right)\;\mathrm{if}\;S^{S}\left(\varphi \right)=+\infty \end{array}\right.$$

Then, the following fundamental inequalities are satisfied [13]
$$0\leq I^{S}\left(\varphi ;\Lambda^{*}\right)\leq S^{S}\left(\varphi \right).$$

In the above discussion, we only mentioned the separable compound state to define the mutual entropy. In the next section, we define the compound states more generally and discuss the formulation of the transmitted complexity and its behavior in quantum dynamical systems based on C*-systems.

## 3. Compound States

Based on [19], we briefly review the constructions of entangled compound states.

For an initial state φ and a quantum channel $\Lambda^{*}$, the compound state Φ satisfies the marginal conditions:$$\begin{array}{c} tr_{2}\Phi =\varphi \;\;\;\left(\mathrm{marginal}\;\mathrm{condition}\;1\right),\\ tr_{1}\Phi =\Lambda^{*}\varphi \;\left(\mathrm{marginal}\;\mathrm{condition}\;2\right). \end{array}$$

We introduced formulations of entangled compound states $\Phi_{E}^{\left(\Delta \right)}$ in [19]. They are not sufficient representations of the compound state. This compound state dose not use the possible decomposition of all states. Therefore, we introduced a new compound state called a modified compound state in [19] and investigated whether the fundamental inequality of mutual entropy held.

For any normal state $\varphi \in S(A_{1})$ of $A_{1}=B\left(H_{1}\right)$, there exists a density operator $\rho \in S\left(H_{1}\right)$ (i.e., the set of all density operators on $H_{1}$) satisfying
$$\varphi \left(A\right)=tr\rho A,\;\left(\forall A\in A_{1}\right),$$
where $H_{1}$ is the Hilbert space of the initial system. Let $\sum_{n\in Q}\lambda_{n}E_{n}$ be a Schatten decomposition [43] of ρ with respect to φ; then, we have
$$\varphi \left(A\right)=tr\sum_{n\in Q}\lambda_{n}E_{n}A=\sum_{n\in Q}\lambda_{n}tr\omega_{n,n}\left(A\right),\;\left(\forall A\in A_{1}\right),$$
where $E_{n}=|x_{n}\rangle \langle x_{n}|$ is the trace class operator associated with $\omega_{n,n}\in S\left(A_{1}\right)$
$$\omega_{n,n}\left(A\right)=trE_{n}A$$
for any $A\in A_{1}.$

(1) The separable compound state ${\tilde{\Phi }}_{\mu ,E,\Lambda^{*}}^{S\left(1\right)}$ of φ and a CP channel $\Lambda^{*}$ from $S(A_{1})$ to $S(A_{2})$ is given by
$${\tilde{\Phi }}_{\mu ,E,\Lambda^{*}}^{S\left(1\right)}=\sum_{n_{k}\in Q}\lambda_{n_{k}}\omega_{n_{k},n_{k}}⊗\Lambda^{*}\left(\omega_{n_{k},n_{k}}\right)$$
for $\mu \in F_{\varphi }^{\varepsilon }\left(S\right)$ of φ.

(2) The full entangled compound state ${\tilde{\Phi }}_{\mu ,E,\Lambda_{t}^{*}}^{S\left(Q\right)}$ of φ and a CP channel $\Lambda^{*}$ is denoted by
$${\tilde{\Phi }}_{\mu ,E,\Lambda^{*}}^{S\left(Q\right)}=\sum_{n_{i}\in Q}\sum_{n_{j}\in Q}\sqrt{\lambda_{n_{i}}}\sqrt{\lambda_{n_{j}}}\omega_{n_{i},n_{j}}⊗\Lambda^{*}\left(\omega_{n_{i},n_{j}}\right)$$
for $\mu \in F_{\varphi }^{\varepsilon }\left(S\right)$ of φ.

### Modified Compound State through Quantum Markov Process

Based on [19], the modified compound state through a quantum Markov process [44] is formulated as follows. Let *Q* be a partition of the index set *Q* of the Schatten–von Neumann decomposition of ρ for the initial state φ such as
$$\begin{array}{ccc} A & = & \left\{A_{j}\subset Q;\;\#\left(A_{j}\right)\neq Ø\;\left(j\in J\right),\right.\\ & & \left.A_{i}\cap A_{j}=Ø\;\left(i\neq j\right),\;Q=\underset{j\in J}{⋃}A_{j}\right\} \end{array}$$

We denote $A_{\Delta }$ and $A_{1}$ by
$$\begin{array}{ccc} A_{\Delta } & = & \left\{A_{j}\in A;\;\#\left(A_{j}\right)\geq 2\;\left(j\in J\right)\right\},\\ A_{1} & = & \left\{A_{j}\in A;\;\#\left(A_{j}\right)=1\;\left(j\in J\right)\right\}, \end{array}$$
$$A=A_{1}\cup A_{\Delta },$$

Let $J_{\ell }$ and $J_{1}$ be
$$\begin{array}{ccc} J_{\ell } & = & \left\{n_{i}\in J;\;\#\left(A_{n_{i}}\right)=\ell \left(\geq 2\right)\right\}\\ J_{1} & = & \left\{n_{i}\in J;\;\#\left(A_{n_{i}}\right)=1\right\} \end{array}$$

Let $\Delta_{\ell }$ and Δ be the subsets of the index set *Q* by
$$\Delta_{\ell }=\underset{n_{i}\in J_{\ell }}{⋃}A_{n_{i}},\;\Delta =\underset{\ell =2}{⋃}\Delta_{\ell },\;Q\setminus \Delta =\underset{n_{i}\in J_{1}}{⋃}A_{n_{i}}$$

(3) The hybrid compound state ${\tilde{\Phi }}_{\mu ,E,\Lambda^{*},\Gamma ,m}^{\gamma (\cdot )}$ of φ and a CP channel $\Lambda^{*}$ is denoted by
$$\begin{array}{ccc} & & {\tilde{\Phi }}_{\mu ,E,\Lambda^{*},\Gamma ,m}^{\gamma (\cdot )}=\sum_{n_{i}\in \Delta }\sum_{n_{j}\in \Delta }\sqrt{\lambda_{n_{i}}}\sqrt{\lambda_{n_{j}}}\\ & & \sum_{i_{1},\ldots ,i_{m}}\sum_{j_{1},\ldots ,j_{m}}\prod_{k=1}^{n}\omega_{n_{i}n_{j}}\left(|x_{i_{k}}\rangle \langle x_{i_{k}}|\right)\prod_{l=1}^{n}\Lambda^{*}\left(\omega_{n_{i}n_{j}}\right)\left(|y_{j_{l}}\rangle \langle y_{j_{l}}|\right)\\ & & e_{i_{1}i_{1}}⊗\cdots ⊗e_{i_{m}i_{m}}⊗e_{j_{1}j_{1}}⊗\cdots ⊗e_{j_{m}j_{m}}\\ & & +\sum_{n_{k}\in Q\setminus \Delta }\lambda_{n_{k}}\sum_{i_{1},\ldots ,i_{m}}\prod_{k=1}^{n}\omega_{n_{k}n_{k}}\left(|x_{i_{k}}\rangle \langle x_{i_{k}}|\right)\sum_{j_{1},\ldots ,j_{m}}\prod_{l=1}^{n}\Lambda^{*}\left(\omega_{n_{k}n_{k}}\right)\left(|y_{j_{k}}\rangle \langle y_{j_{k}}|\right)\\ & & e_{i_{1}i_{1}}⊗\cdots ⊗e_{i_{m}i_{m}}⊗e_{j_{1}j_{1}}⊗\cdots ⊗e_{j_{m}j_{m}}. \end{array}$$

This compound state does not use the possible decomposition of all states. Thus, we introduce the modified compound state. We put
$$A_{\Delta }=\underset{\ell =2}{⋃}A_{\Delta_{\ell }},$$
where $A_{\Delta_{\ell }}$ is the subset of $A_{\Delta }$ such that
$$A_{\Delta_{\ell }}=\left\{A_{j}\in A;\;\#\left(A_{j}\right)=\ell \;\left(j\in J\right)\right\}\subset A_{\Delta },$$

Let $P\left(Q\right)$ be the set of all partitions of the total index set *Q* by
$$\begin{array}{ccc} P\left(Q\right) & = & \left\{A;A=A_{1}\cup A_{\Delta },\forall A_{i},A_{j}\left(\neq Ø\right)\in A\;\left(i\neq j\right),\right.\\ & & \left.A_{i}\cap A_{j}=Ø,\;Q=\underset{j\in J}{⋃}A_{j}=\underset{\ell =2}{⋃}\left(\underset{n_{i}\in J_{\ell }}{⋃}A_{n_{i}}\right)\cup \left(\underset{n_{i}\in J_{1}}{⋃}A_{n_{i}}\right)\right\} \end{array}$$

By using the trace class operator $|x_{n_{i}}\rangle \langle x_{n_{i}^{\prime }}|$ on $H_{1}$ and ($\sum_{n_{i}\in A_{n_{\ell }}}\sqrt{\lambda_{n_{i}}}|x_{n_{i}}\rangle$)($\sum_{n_{j}\in A_{n_{\ell }}}\sqrt{\lambda_{n_{j}}}\langle x_{n_{j}}|$)∈$S\left(H_{1}\right)$, where $\left\{|x_{n_{i}}\rangle \right\}$ is a CONS (complete orthogonal systems) in H, we express a linear functional $\omega_{n_{i},n_{i}^{\prime }}$ on $A_{1}$ and $\omega_{A_{n_{\ell }},A_{n_{\ell }}}\in S\left(A_{1}\right)$ by
$$\omega_{n_{i},n_{i}^{\prime }}\left(A\right)=tr|x_{n_{i}}\rangle \langle x_{n_{i}^{\prime }}|A$$
for any $A\in A_{1}.$

Let $\Lambda^{*}$ be the CP channel $\Lambda^{*}$ given by $\Lambda^{*}\left(\varphi \right)\left(\cdot \right)=\varphi \left(V^{*}\left(\cdot \right)V\right)$ for any φ, satisfying $V^{*}V=I$ and $\Lambda^{*}\left(\omega_{nm}\right)\left(A\right)=\delta_{nm}\omega_{nm}\left(V^{*}\left(A\right)V\right)$ for any $A\in A_{1}$.

Based on the Jamiolkowski isomorphic channel [27], the modified compound states is defined as follows.

(4) The modified compound state ${\tilde{\Phi }}_{\mu ,E,\Lambda^{*},\Gamma ,m}^{\gamma (A)}$ by means of partitions A of the total index set *Q* with respect to the Schatten decomposition of ρ of the initial state φ and the CP channel $\Lambda^{*}$ is given by
$$\begin{array}{ccc} & & {\tilde{\Phi }}_{\mu ,E,\Lambda^{*},\Gamma ,m}^{\gamma (A)}=\sum_{\ell =2}\sum_{n_{\ell }\in J_{\ell }}\sum_{n_{i}\in A_{n_{\ell }}}\sum_{n_{j}\in A_{n_{\ell }}}\sqrt{\lambda_{n_{i}}}\sqrt{\lambda_{n_{j}}}\\ & & \sum_{i_{1},\ldots ,i_{m}}\sum_{j_{1},\ldots ,j_{m}}\prod_{k=1}^{n}\omega_{n_{i}n_{j}}\left(|x_{i_{k}}\rangle \langle x_{i_{k}}|\right)\prod_{l=1}^{n}\Lambda^{*}\left(\omega_{n_{i}n_{j}}\right)\left(|y_{j_{l}}\rangle \langle y_{j_{l}}|\right)\\ & & e_{i_{1}i_{1}}⊗\cdots ⊗e_{i_{m}i_{m}}⊗e_{j_{1}j_{1}}⊗\cdots ⊗e_{j_{m}j_{m}}\\ & & +\sum_{n_{k}\in Q\setminus \Delta }\lambda_{n_{k}}\sum_{i_{1},\ldots ,i_{m}}\prod_{k=1}^{n}\omega_{n_{k}n_{k}}\left(|x_{i_{k}}\rangle \langle x_{i_{k}}|\right)\sum_{j_{1},\ldots ,j_{m}}\prod_{l=1}^{n}\Lambda^{*}\left(\omega_{n_{k}n_{k}}\right)\left(|y_{j_{k}}\rangle \langle y_{j_{k}}|\right)\\ & & e_{i_{1}i_{1}}⊗\cdots ⊗e_{i_{m}i_{m}}⊗e_{j_{1}j_{1}}⊗\cdots ⊗e_{j_{m}j_{m}}. \end{array}$$

## 4. Transmitted Complexity for the Modified Compound States in Dynamical Systems

We discussed the transmitted complexity for the modified compound states in dynamical systems described by the Hilbert spaces [40]. In this paper, we discuss these problems on the C*-systems.

We here define the bijection $\Xi_{1}$ from $S(A_{1})$ to $S\left(H_{1}\right)$ by
$$\Xi_{1}\left(\varphi \right)=\rho ,\;\varphi \left(A\right)=tr\Xi_{1}\left(\varphi \right)A$$
for any density operator $\rho \in S\left(H_{1}\right)$.

Let $M_{d}$ (respectively, $M_{d}^{\prime }$) be the set of all d×d matrices of an input and output systems, respectively.

We use the state $\Xi_{1}{\left(\varphi \right)}_{\Gamma ,m}$ on ${⊗}_{1}^{m}M_{d}$. Then,
$$\Xi_{1}{\left(\varphi \right)}_{\Gamma ,m}=\sum_{i_{1}\cdots i_{m}}\prod_{k=1}^{n}\langle x_{i_{k}},\Xi_{1}\left(\varphi \right)x_{i_{k}}\rangle e_{i_{1}i_{1}}⊗\ldots ⊗e_{i_{m}i_{m}},$$
where $e_{kk}$ is diagonal elements of $M_{d}.$ Assume that $\Xi_{1}\left(\varphi \right)$ is a density operator on $H_{1}$, then we have the state $\Xi_{1}{\left(\varphi \right)}_{\Gamma^{\prime },m}^{\Lambda^{*}}\in S\left({⊗}_{1}^{m}M_{d}^{\prime }\right)$ as
$$\Xi_{1}{\left(\varphi \right)}_{\Gamma^{\prime },m}^{\Lambda^{*}}=\sum_{j_{1}\cdots j_{m}}\prod_{l=1}^{n}\langle y_{j_{l}},\Lambda^{*}\left(\Xi_{1}\left(\varphi \right)\right)y_{j_{l}}\rangle e_{j_{1}j_{1}}^{\prime }⊗\ldots ⊗e_{j_{m}j_{m}}^{\prime }$$
where $e_{j_{k}j_{k}}^{\prime }\in S\left(M_{d}^{\prime }\right).$ For the initial state $\varphi \in S\left(A_{1}\right)$, the generalized AOW entropy $S_{m}^{\gamma (A)}\left(\varphi ;\gamma ,\theta \right)$ with respect to γ,θ and *m* is defined by
$$S_{m}^{\gamma (A)}\left(\varphi ;\gamma ,\theta \right)=\;S\left(\Xi_{1}{\left(\varphi \right)}_{\Gamma ,m}\right).$$

For the initial state $\varphi \in S\left(A_{1}\right)$, the generalized AOW entropy ${\tilde{S}}_{m}^{\gamma (A)}\left(\varphi ;\gamma ,\theta \right)$ with respect to γ,θ is defined by
$${\tilde{S}}^{\gamma (A)}\left(\varphi ;\gamma ,\theta \right)=\underset{m\to \infty }{\lim\sup}\frac{1}{m}S_{m}^{\gamma (A)}\left(\varphi ;\gamma ,\theta \right).$$

Then, the trivial compound state through quantum Markov chains is given by
$$\begin{array}{ccc} \Phi_{0}^{\left(m\right)} & = & \Xi_{1}{\left(\varphi \right)}_{\Gamma ,m}⊗\Xi_{1}{\left(\varphi \right)}_{\Gamma^{\prime },m}^{\Lambda^{*}}\\ & = & \sum_{i_{1}\ldots i_{m}}\sum_{j_{1}\ldots j_{m}}\prod_{k=1}^{n}\langle x_{i_{k}},\Xi_{1}\left(\varphi \right)x_{i_{k}}\rangle \prod_{l=1}^{n}\langle y_{j_{l}},\Lambda^{*}\left(\Xi_{1}\left(\varphi \right)\right)y_{j_{l}}\rangle \\ & \; & \left(e_{i_{1}i_{1}}⊗\ldots ⊗e_{i_{m}i_{m}}\right)⊗\left(e_{j_{1}j_{1}}^{\prime }⊗\ldots ⊗e_{j_{m}j_{m}}^{\prime }\right) \end{array}$$

For the modified compound state ${\tilde{\Phi }}_{\mu ,E,\Lambda^{*},\Gamma ,m}^{\gamma (A)}$, the transmitted complexity $I_{m}^{\gamma (\Delta )}(\varphi$; $\Lambda^{*}$, γ, $\gamma^{\prime }$, θ, $\theta^{\prime }$) with respect to $\Lambda^{*}$, γ, $\gamma^{\prime }$, θ, $\theta^{\prime }$ and *m* is defined by
$$I_{m}^{\gamma (A)}\left(\varphi ;\Lambda^{*},\gamma ,\gamma^{\prime },\theta ,\theta^{\prime }\right)=\underset{E}{\sup}S\left({\tilde{\Phi }}_{\mu ,E,\Lambda^{*},\Gamma ,m}^{\gamma (A)},\Phi_{0}^{\left(m\right)}\right)$$

**Definition 1.** 
*The quantum dynamical mutual entropy for the modified compound state ${\tilde{\Phi }}_{\mu ,E,\Lambda^{*},\Gamma ,m}^{\gamma (A)}$ through quantum Markov chains with respect to $\rho ,\Lambda^{*},\gamma ,\gamma^{\prime },\theta ,\theta^{\prime }$ and the decomposition of φ is defined by*

(5)
$${\tilde{I}}^{\gamma (A)}\left(\varphi ;\Lambda^{*},\gamma ,\gamma^{\prime },\theta ,\theta^{\prime }\right)=\underset{m\to \infty }{lim sup}\frac{1}{m}I_{m}^{\gamma (A)}\left(\varphi ;\Lambda^{*},\gamma ,\gamma^{\prime },\theta ,\theta^{\prime }\right).$$



Then, we have:

**Theorem 1.** 
*Let $\sum_{n\in Q}\lambda_{n}E_{n}$ be a Schatten decomposition of $\Xi_{1}\left(\varphi \right)$. For the modified compound state ${\tilde{\Phi }}_{\mu ,E,\Lambda^{*},\Gamma ,m}^{\gamma (A)}$ with respect to a partition A of the index set Q and the CP channel $\Lambda^{*}\left(\Xi_{1}\left(\varphi \right)\right)=V\left(\Xi_{1}\left(\varphi \right)\right)V^{*}$ for any $\Xi_{1}\left(\varphi \right)\in S\left(H_{1}\right)$ with $V^{*}V=I$, two marginal conditions hold*

$$tr_{j_{1},\ldots ,j_{m}}{\tilde{\Phi }}_{\mu ,E,\Lambda^{*},\Gamma ,m}^{\gamma (A)}=\Xi_{1}\left(\varphi \right){}_{\Gamma ,m}\;and\;\;tr_{i_{1},\cdots ,i_{m}}{\tilde{\Phi }}_{\mu ,E,\Lambda^{*},\Gamma ,m}^{\gamma (A)}=\Xi_{1}{\left(\varphi \right)}_{\Gamma^{\prime },m}^{\Lambda^{*}}$$


*and the transmitted complexity with respect to ${\tilde{\Phi }}_{\mu ,E,\Lambda^{*},\Gamma ,m}^{\gamma (A)}$ and $\Phi_{0}^{\left(m\right)}=\Xi_{1}{\left(\varphi \right)}_{\Gamma ,m}⊗\Xi_{1}{\left(\varphi \right)}_{\Gamma^{\prime },m}^{\Lambda^{*}}$ fulfills the fundamental inequalities:*

$$0\leq {\tilde{I}}^{\gamma \left(A\right)}\left(\varphi ;\Lambda^{*},\gamma ,\gamma^{\prime },\theta ,\theta^{\prime }\right)\leq \min\left\{{\tilde{S}}^{\gamma (A)}\left(\varphi ;\gamma ,\theta \right),{\tilde{S}}^{\gamma (A)}\left(\varphi ;\Lambda^{*},\gamma^{\prime },\theta^{\prime }\right)\right\}.$$



**Proof.** Applying the partial traces for ${⊗}_{1}^{m}K_{2}$ and for ${⊗}_{1}^{m}K_{1}$ with respect to ${\tilde{\Phi }}_{\mu ,E,\Lambda^{*},\Gamma ,m}^{\gamma (A)}$, ${\tilde{\Phi }}_{\mu ,E,\Lambda^{*},\Gamma ,m}^{\gamma (A)}$ holds two marginal conditions.
$$\begin{array}{ccc} & & tr_{\left({⊗}_{1}^{m}K_{2}\right)}{\tilde{\Phi }}_{\mu ,E,\Lambda^{*},\Gamma ,m}^{\gamma (A)}\\ & = & \sum_{i_{1},\ldots ,i_{m}}tr\left(\Xi_{1}\left(\varphi \right){\left|\Gamma_{i_{m}\ldots i_{1}}\right|}^{2}\right)e_{i_{1}i_{1}}⊗\cdots ⊗e_{i_{m}i_{m}}\\ & = & \Xi_{1}{\left(\varphi \right)}_{\Gamma ,m} \end{array}$$
and
$$\begin{array}{ccc} & & tr_{\left({⊗}_{1}^{m}K_{1}\right)}{\tilde{\Phi }}_{\mu ,E,\Lambda^{*},\Gamma ,m}^{\gamma (A)}\\ & = & \sum_{j_{1},\ldots ,j_{m}}tr\left[\Lambda^{*}\left(\Xi_{1}\left(\varphi \right)\right){\left|\Gamma_{j_{m}j_{m-1}\ldots j_{1}}^{\prime }\right|}^{2}\right]e_{j_{1}j_{1}}⊗\cdots ⊗e_{j_{m}j_{m}}\\ & = & \Xi_{1}{\left(\varphi \right)}_{\Gamma ,m}^{\Lambda^{*}} \end{array}$$$S\left({\tilde{\Phi }}_{\mu ,E,\Lambda^{*},\Gamma ,m}^{\gamma (A)},\Phi_{0}^{\left(m\right)}\right)$ is denoted by
$$S\left({\tilde{\Phi }}_{\mu ,E,\Lambda^{*},\Gamma ,m}^{\gamma (A)},\Phi_{0}^{\left(m\right)}\right)=tr{\tilde{\Phi }}_{\mu ,E,\Lambda^{*},\Gamma ,m}^{\gamma (A)}\log{\tilde{\Phi }}_{\mu ,E,\Lambda^{*},\Gamma ,m}^{\gamma (A)}-tr{\tilde{\Phi }}_{\mu ,E,\Lambda^{*},\Gamma ,m}^{\gamma (A)}\log\Phi_{0}^{\left(m\right)}$$The second term is written as
$$tr{\tilde{\Phi }}_{\mu ,E,\Lambda^{*},\Gamma ,m}^{\gamma (A)}\log\left(\Xi_{1}{\left(\varphi \right)}_{\Gamma ,m}⊗\Xi_{1}{\left(\varphi \right)}_{\Gamma ,m}^{\Lambda^{*}}\right)=tr\Xi_{1}{\left(\varphi \right)}_{\Gamma ,m}\log\Xi_{1}{\left(\varphi \right)}_{\Gamma ,m}+tr\Xi_{1}{\left(\varphi \right)}_{\Gamma ,m}^{\Lambda^{*}}\log\Xi_{1}{\left(\varphi \right)}_{\Gamma ,m}^{\Lambda^{*}}$$The first term is described by
$$\begin{array}{ccc} & & tr{\tilde{\Phi }}_{\mu ,E,\Lambda^{*},\Gamma ,m}^{\gamma (A)}\log{\tilde{\Phi }}_{\mu ,E,\Lambda^{*},\Gamma ,m}^{\gamma (A)}\\ & = & \sum_{i_{1},\ldots ,i_{m}}\sum_{j_{1},\ldots ,j_{m}}\left[\sum_{\ell =2}\sum_{n_{\ell }\in J_{\ell }}\right.\\ & & \sum_{n_{i}\in A_{n_{\ell }}}\sum_{n_{j}\in A_{n_{\ell }}}\sqrt{\lambda_{n_{i}}}\sqrt{\lambda_{n_{j}}}\prod_{k=1}^{n}\omega_{n_{i}n_{j}}\left(|x_{i_{k}}\rangle \langle x_{i_{k}}|\right)\prod_{l=1}^{n}\Lambda^{*}\left(\omega_{n_{i}n_{j}}\right)\left(|y_{j_{k}}\rangle \langle y_{j_{k}}|\right)\\ & & \left.+\sum_{n_{k}\in Q\setminus \Delta }\lambda_{n_{k}}\prod_{k=1}^{n}\omega_{n_{k}n_{k}}\left(|x_{i_{k}}\rangle \langle x_{i_{k}}|\right)\prod_{l=1}^{n}\Lambda^{*}\left(\omega_{n_{k}n_{k}}\right)\left(|y_{j_{k}}\rangle \langle y_{j_{k}}|\right)\right]\log\\ & & \left[\sum_{\ell =2}\sum_{n_{\ell }\in J_{\ell }}\right.\\ & & \sum_{n_{i}\in A_{n_{\ell }}}\sum_{n_{j}\in A_{n_{\ell }}}\sqrt{\lambda_{n_{i}}}\sqrt{\lambda_{n_{j}}}\prod_{k=1}^{n}\omega_{n_{i}n_{j}}\left(|x_{i_{k}}\rangle \langle x_{i_{k}}|\right)\prod_{l=1}^{n}\Lambda^{*}\left(\omega_{n_{i}n_{j}}\right)\left(|y_{j_{k}}\rangle \langle y_{j_{k}}|\right)\\ & & \left.+\sum_{n_{k}\in Q\setminus \Delta }\lambda_{n_{k}}\prod_{k=1}^{n}\omega_{n_{k}n_{k}}\left(|x_{i_{k}}\rangle \langle x_{i_{k}}|\right)\prod_{l=1}^{n}\Lambda^{*}\left(\omega_{n_{k}n_{k}}\right)\left(|y_{j_{k}}\rangle \langle y_{j_{k}}|\right)\right] \end{array}$$Since $S\left(\Xi_{1}{\left(\varphi \right)}_{\Gamma ,m}\right)$ is written by
$$\begin{array}{ccc} & & S\left(\Xi_{1}{\left(\varphi \right)}_{\Gamma ,m}\right)\\ & = & -\sum_{i_{1},\ldots ,i_{m}}\sum_{j_{1},\ldots ,j_{m}}\left[\sum_{\ell =2}\sum_{n_{\ell }\in J_{\ell }}\right.\\ & & \sum_{n_{i}\in A_{n_{\ell }}}\sum_{n_{j}\in A_{n_{\ell }}}\sqrt{\lambda_{n_{i}}}\sqrt{\lambda_{n_{j}}}\prod_{k=1}^{n}\omega_{n_{i}n_{j}}\left(|x_{i_{k}}\rangle \langle x_{i_{k}}|\right)\prod_{l=1}^{n}\Lambda^{*}\left(\omega_{n_{i}n_{j}}\right)\left(|y_{j_{k}}\rangle \langle y_{j_{k}}|\right)\\ & & \left.+\sum_{n_{k}\in Q\setminus \Delta }\lambda_{n_{k}}\prod_{k=1}^{n}\omega_{n_{k}n_{k}}\left(|x_{i_{k}}\rangle \langle x_{i_{k}}|\right)\prod_{l=1}^{n}\Lambda^{*}\left(\omega_{n_{k}n_{k}}\right)\left(|y_{j_{k}}\rangle \langle y_{j_{k}}|\right)\right]\log\\ & & \sum_{n_{k}\in Q}\lambda_{n_{k}}\prod_{k=1}^{n}\omega_{n_{k}n_{k}}\left(|x_{i_{k}}\rangle \langle x_{i_{k}}|\right), \end{array}$$
then one can obtain
$$\begin{array}{ccc} & & S\left({\tilde{\Phi }}_{\mu ,E,\Lambda^{*},\Gamma ,m}^{\gamma (A)},\Phi_{0}^{\left(m\right)}\right)\\ & = & tr{\tilde{\Phi }}_{\mu ,E,\Lambda^{*},\Gamma ,m}^{\gamma (A)}\log{\tilde{\Phi }}_{\mu ,E,\Lambda^{*},\Gamma ,m}^{\gamma (A)}+S\left(\Xi_{1}{\left(\varphi \right)}_{\Gamma ,m}\right)+S\left(\Xi_{1}{\left(\varphi \right)}_{\Gamma ,m}^{\Lambda^{*}}\right)\\ & = & S\left(\Xi_{1}{\left(\varphi \right)}_{\Gamma ,m}^{\Lambda^{*}}\right)+\sum_{i_{1},\ldots ,i_{m}}\sum_{j_{1},\ldots ,j_{m}}\left[\sum_{\ell =2}\sum_{n_{\ell }\in J_{\ell }}\right.\\ & & \sum_{n_{i}\in A_{n_{\ell }}}\sum_{n_{j}\in A_{n_{\ell }}}\sqrt{\lambda_{n_{i}}}\sqrt{\lambda_{n_{j}}}\prod_{k=1}^{n}\omega_{n_{i}n_{j}}\left(|x_{i_{k}}\rangle \langle x_{i_{k}}|\right)\prod_{l=1}^{n}\Lambda^{*}\left(\omega_{n_{i}n_{j}}\right)\left(|y_{j_{k}}\rangle \langle y_{j_{k}}|\right)\\ & & \left.+\sum_{n_{k}\in Q\setminus \Delta }\lambda_{n_{k}}\prod_{k=1}^{n}\omega_{n_{k}n_{k}}\left(|x_{i_{k}}\rangle \langle x_{i_{k}}|\right)\prod_{l=1}^{n}\Lambda^{*}\left(\omega_{n_{k}n_{k}}\right)\left(|y_{j_{k}}\rangle \langle y_{j_{k}}|\right)\right]\\ & & \log\frac{\Theta_{i_{m},\cdots ,i_{1},j_{m},\cdots ,j_{1}}}{\sum_{j_{1},\ldots ,j_{m}}\Theta_{i_{m},\cdots ,i_{1},j_{m},\cdots ,j_{1}}}\\ & \leq & S\left(\Xi_{1}{\left(\varphi \right)}_{\Gamma ,m}^{\Lambda^{*}}\right), \end{array}$$
where
$$\begin{array}{ccc} & & \Theta_{i_{m},\cdots ,i_{1},j_{m},\cdots ,j_{1}}\\ & = & \sum_{\ell =2}\sum_{n_{\ell }\in J_{\ell }}\sum_{n_{i}\in A_{n_{\ell }}}\sum_{n_{j}\in A_{n_{\ell }}}\sqrt{\lambda_{n_{i}}}\sqrt{\lambda_{n_{j}}}\prod_{k=1}^{n}\omega_{n_{i}n_{j}}\left(|x_{i_{k}}\rangle \langle x_{i_{k}}|\right)\prod_{l=1}^{n}\Lambda^{*}\left(\omega_{n_{i}n_{j}}\right)\left(|y_{j_{k}}\rangle \langle y_{j_{k}}|\right)\\ & & +\sum_{n_{k}\in Q\setminus \Delta }\lambda_{n_{k}}\prod_{k=1}^{n}\omega_{n_{k}n_{k}}\left(|x_{i_{k}}\rangle \langle x_{i_{k}}|\right)\prod_{l=1}^{n}\Lambda^{*}\left(\omega_{n_{k}n_{k}}\right)\left(|y_{j_{k}}\rangle \langle y_{j_{k}}|\right). \end{array}$$Since $S\left(\Xi_{1}{\left(\varphi \right)}_{\Gamma^{\prime },m}^{\Lambda^{*}}\right)$ is described by
$$\begin{array}{ccc} & & S\left(\Xi_{1}{\left(\varphi \right)}_{\Gamma^{\prime },m}^{\Lambda^{*}}\right)\\ & = & -\sum_{i_{1},\ldots ,i_{m}}\sum_{j_{1},\ldots ,j_{m}}\left[\sum_{\ell =2}\sum_{n_{\ell }\in J_{\ell }}\right.\\ & & \sum_{n_{i}\in A_{n_{\ell }}}\sum_{n_{j}\in A_{n_{\ell }}}\sqrt{\lambda_{n_{i}}}\sqrt{\lambda_{n_{j}}}\prod_{k=1}^{n}\omega_{n_{i}n_{j}}\left(|x_{i_{k}}\rangle \langle x_{i_{k}}|\right)\prod_{l=1}^{n}\Lambda^{*}\left(\omega_{n_{i}n_{j}}\right)\left(|y_{j_{k}}\rangle \langle y_{j_{k}}|\right)\\ & & \left.+\sum_{n_{k}\in Q\setminus \Delta }\lambda_{n_{k}}\prod_{k=1}^{n}\omega_{n_{k}n_{k}}\left(|x_{i_{k}}\rangle \langle x_{i_{k}}|\right)\prod_{l=1}^{n}\Lambda^{*}\left(\omega_{n_{k}n_{k}}\right)\left(|y_{j_{k}}\rangle \langle y_{j_{k}}|\right)\right]\log\\ & & \sum_{n_{k}\in Q}\lambda_{n_{k}}\prod_{k=1}^{n}\omega_{n_{k}n_{k}}\left(|x_{i_{k}}\rangle \langle x_{i_{k}}|\right) \end{array}$$
then we have
$$\begin{array}{ccc} & & S\left({\tilde{\Phi }}_{\mu ,E,\Lambda^{*},\Gamma ,m}^{\gamma (A)},\Phi_{0}^{\left(m\right)}\right)\\ & = & tr{\tilde{\Phi }}_{\mu ,E,\Lambda^{*},\Gamma ,m}^{\gamma (A)}\log{\tilde{\Phi }}_{\mu ,E,\Lambda^{*},\Gamma ,m}^{\gamma (A)}+S\left(\Xi_{1}{\left(\varphi \right)}_{\Gamma ,m}\right)+S\left(\Xi_{1}{\left(\varphi \right)}_{\Gamma^{\prime },m}^{\Lambda^{*}}\right)\\ & = & S\left(\rho_{\Gamma ,m}\right)+\sum_{i_{1},\ldots ,i_{m}}\sum_{j_{1},\ldots ,j_{m}}\left[\sum_{\ell =2}\sum_{n_{\ell }\in J_{\ell }}\right.\\ & & \sum_{n_{i}\in A_{n_{\ell }}}\sum_{n_{j}\in A_{n_{\ell }}}\sqrt{\lambda_{n_{i}}}\sqrt{\lambda_{n_{j}}}\prod_{k=1}^{n}\omega_{n_{i}n_{j}}\left(|x_{i_{k}}\rangle \langle x_{i_{k}}|\right)\prod_{l=1}^{n}\Lambda^{*}\left(\omega_{n_{i}n_{j}}\right)\left(|y_{j_{k}}\rangle \langle y_{j_{k}}|\right)\\ & & \left.+\sum_{n_{k}\in Q\setminus \Delta }\lambda_{n_{k}}\prod_{k=1}^{n}\omega_{n_{k}n_{k}}\left(|x_{i_{k}}\rangle \langle x_{i_{k}}|\right)\prod_{l=1}^{n}\Lambda^{*}\left(\omega_{n_{k}n_{k}}\right)\left(|y_{j_{k}}\rangle \langle y_{j_{k}}|\right)\right]\\ & & \log\frac{\Theta_{i_{m},\cdots ,i_{1},j_{m},\cdots ,j_{1}}}{\sum_{i_{1},\ldots ,i_{m}}\Theta_{i_{m},\cdots ,i_{1},j_{m},\cdots ,j_{1}}}\\ & \leq & S\left(\Xi_{1}{\left(\varphi \right)}_{\Gamma ,m}\right) \end{array}$$Therefore, we get the following inequality:
$$0\leq S\left({\tilde{\Phi }}_{\mu ,E,\Lambda^{*},\Gamma ,m}^{\gamma (A)},\Phi_{0}^{\left(m\right)}\right)\leq \min\left\{S\left(\rho_{\Gamma ,m}\right),S\left(\rho_{\Gamma^{\prime },m}^{\Lambda^{*}}\right)\right\}.$$Applying the supremum of *E* of both sides of the above inequalities, one has
$$0\leq I_{m}^{\gamma (A)}\left(\varphi ;\Lambda^{*},\gamma ,\gamma^{\prime },\theta ,\theta^{\prime }\right)\leq \min\left\{S_{m}^{\gamma (A)}\left(\varphi ;\gamma ,\theta \right),S_{m}^{\gamma (A)}\left(\varphi ;\Lambda^{*},\gamma^{\prime },\theta^{\prime }\right)\right\}.$$Thus, we have the inequalities taking a ${lim sup}_{m\to \infty }\frac{1}{m}$ of both sides of the above inequalities:
$$0\leq I^{\gamma (A)}\left(\varphi ;\Lambda^{*},\gamma ,\gamma^{\prime },\theta ,\theta^{\prime }\right)\leq \min\left\{{\tilde{S}}^{\gamma (A)}\left(\varphi ;\gamma ,\theta \right),{\tilde{S}}^{\gamma (A)}\left(\varphi ;\Lambda^{*},\gamma^{\prime },\theta^{\prime }\right)\right\}.$$□

## 5. Conclusions

Ohya’s quantum mutual entropy for CP channels and quantum communication processes was shown to be effective in C* dynamical systems. The transmitted complexity for the modified compound states in dynamical systems described by the Hilbert spaces was discussed in [40]. In this paper, we discussed these problems on the C*-systems. Based on the generalized AOW entropy formulated by the KOW entropy in the C* dynamical system, we investigated the complexity associated with the entangled compound states. It was shown that the fundamental inequalities were satisfied when the mutual entropy of the initial state transmitted through the CP channel changed with time (steps *m*). Note, however, that this result does not assert the validity of the modified compound state given by the entangled state, since the efficiency of the information transmission of the initial state alone decreases with time. This means that the inequalities of the complexities based on the modified compound state are not satisfied in all cases. For example, their inequalities are not satisfied at the initial situation (see [19]).

## Data Availability

No new data were created or analyzed in this study. Data sharing is not applicable to this article.

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
