# Peer review of "On Transmitted Complexity Based on Modified Compound States"

_entropy, 2023, doi:10.3390/e25030455_

Round 1

Reviewer 1 Report

The manuscript entropy-2153290, which has been submitted for publication to Entropy, elaborates on the concepts of S-mixing entropy in general quantum systems and the quantum dynamical entropy. The latter was proposed in 1999 by the author of the present manuscript along with two eminent researchers, i.e., M. Ohya and A. Kossakowski. Consequently, the author refers to this concepts as KOW entropy.

The paper discusses the use of Ohya's quantum mutual entropy in C* dynamical systems for CP channels and quantum communication processes. It explores the complexity associated with entangled compound states and shows that the fundamental inequalities are satisfied when the mutual entropy of the initial state transmitted through the CP channel changes with time. However, the efficiency of information transmission of the initial state alone decreases with time, and the validity of the modified compound state given by the entangled state is not asserted.

I consider the paper well-suited to the journal Entropy. The content of the paper is presented rigorously, according to the standards of mathematical physics. I think the manuscript can be published in Entropy after a minor correction.

My specific comments for the author are listed below.

I. Please rewrite the abstract so that it clearly provides the goals of the paper. Now, in my opinion, it gives to much background information. It should clearly communicate the research methods, design, major findings, and the conclusions reached.

II. Section 2. "Quantum Channels" seems very brief. It might be a good idea to merge Sections 2 & 3 into one.

III. Please ensure that each abbreviation is explained in the manuscript. For example, I do not find any explanation for "AOW." I suppose this acronym comes from the names: Accardi, Ohya and Watanabe (concept introduced in Ref. [3]).

IV. Please read carefully through the manuscript to detect and correct all minor errors. For example, in line 45, we see a question mark instead of a relevant reference. Also, please improve any language problems; for example, line 133 reads: "We The transmitted complexity for," which is apparently a kind of error. Finally, ensure that the references are written in a consistent manner.

Author Response

Dear Sir

Thank you very much for your useful comments.

We have revised my manuscrip according to your suggestions.

Reviewer 2 Report

The paper is devoted to the entropic measures of complexity for quantum channels. Namely, the ideas of Ohya about the entropy of a joint input-output state are developed, a theorem is proved. Unfortunately, I was not able to understand which problem is solved by this paper and theorem. Introduction contains a review of this direction of research, but no one problem was formulated. Both title, abstract and introduction stress that the concept of modified compound (input-output) state is used. But it is not explain why do we need a modification, why the original (non-modified) compound states are not enough.

So, a theorem is formulated, which is, probably, true, but this theorem is very cumbersome and I could not understand its meaning and significance. Besides, the presentation is not clear since many notations are not explained, see comments below.

The paper can be recommended for publication whenever the author explain what was the problem solved in this paper and takes into account the comments given below:

1. The abbreviation KOW is explained in the abstract, but the abbreviations AOW and AF from abstract, introduction and conclusions are not explained. Moreover, we see all these three abbreviations only in the mentioned sections. We do not see them in the main part of the text. Where are formulas for KOW, AOW and AF entropies in the text?

2. Conclusions, lines 137-139: "It is shown that the fundamental inequalities are satisfied when the mutual entropy of the initial state transmitted through the CP channel changes with time." The time is mentioned here (in conclusions!) for the first time. But there is no time in the main part of the paper. Is an action of a quantum channel meant here by "time"?

3. Lines 139-141: "Note, however, that this result does not assert the validity of the modified compound state given by the entangled state, since the efficiency of the information transmission of the initial state alone decreases with time." This sentence is not clear. What does validity of a compound state mean?

4. 2nd line of the section "2. Quantum Channels": what is \mathfrak S(A_1) mathematically. It is written that it is a quantum system, but it is better to write mathematically that it is the set of states (i.e., positive functional preserving the unity) on A_1. Moreover, in the same line, it is written that a quantum system is the whole pair (A_1, S(A_1)), not S(A_1) alone. Moreover, in line 69, another definition of  \mathfrak S(A) is given. It is stated that it is the set of all normal states on A.

5. 4th line of the section "2. Quantum Channels": What is the identity operator in A_k if A_k is an abstract C*-algebra (it is not written that it is an algebra of operators in a Hilbert space).

6. Line 70: Though one can guess what is ex S, an explanation of this notation should be given.

7. Line 79: Is it true that S^{\mathfrak S(A)}(phi) is the von Neumann entropy of phi? If yes, it is better to stress it for convenience of the reader.

8. Lines 81 and 82: "The compound state is defined as", but then two different formulas (Phi0 and PhiSmu) are given. So, not a single compound state is defined, but two compound states are defined?

9. Last lines of p. 3: what are H1 and B(H1). Also, what is \mathfrak S(H_1)? Previously, only \mathfrak S(A), where A is a C*-algebra, was defined.

10. 5th line above Section "4. Compound states": The formula S(PhiSmu,Phi0) should be given for self-consistency of the paper.

11. 2nd line above Section "4. Compound states": "Then the following fundamental inequalities are satisfied". A reference is required.

12. After line 90, it is hard to understand and more explanations are required. For example, what is the meaning of the states Psi and what is the difference between these compound states and the compound states Phiand PhiSmu defined above? Where is the dependence of the states Psi on phi?

13. After line 108, yet another compound state is defined. What is their meaning in comparison with the compound states Phiand PhiSmu and with the compound states Psi defined after lines 90 and 91? In lines 87-88, it is written that the Phi states "are not sufficient representations of the compound state". Why?

14. What are e in the formula after line 108 and in the last line of p. 5? What are |x_ni> between lines 100 and 101 (say, are they elements of an orthonormal basis of the Hilbert space H1?) and in the last line of p.5?

15. 1st line of p. 6: What is a stationary state of H1? This notions was not defined.

16. Between lines 118 and 119: What is "the modified compound state"? In contrast to this state, the state Psi defined after line 108 does not have mu among its subscripts.

17. In Theorem: What is \widetilde S? I cannot find a definition.

Minor typos and language issues:

1. I cannot understand the very end of the abstract: "when the mutual entropy when the initial state is transmitted over the channel changes with time" (the 1st "when" is unclear).

2. Line 69: "has"->"there is".

3. Lines 71-72: It is hard to understand the sentence beginning with "Let mu be...". Probably, it should be splitted into two sentences: "Let my be A [I think, not THE, since it is not unique] measure satisfying the above decomposition. It is not unique unless S is a Choquet simplex".

4. Probably, S in lines 77 and 78 should be calligraphic (\mathcal S) since it refers to the superscript of S in the left-hand side of Eq. (4). 

5. In the formulas after line 98 and line 99, different kinds of the symbol # are used.

6. Formula above line 116: In the last factor, we see "!" instead of "e".

7. Line 45: something wrong with the references (mark "?")

8. Line 133: probably, "We" before "The" should be removed.

Author Response

(The authors gave the same response as above.)

Round 2

Reviewer 1 Report

The manuscript has been amended according to my first report. In my opinion, it can be published in the present form.

Author Response

Dear Sir

Thank you very much for your acceptance of my manuscript.

Reviewer 2 Report

The author has addressed the comments and made the corresponding changes. Now the meaning of the results are a bit clearer (though not entirely clear for me) and the paper can be recommended for publication. I suggest the following further improvements to the author:

P.2, 2nd line from bottom: "We introduced hybrid compound state. This compound state did not use the possible decomposition of all states".

1. Probably, "did not" should be replaced by "does not". 

2. The second sentence should explain the new notion of the hybrid compound state. I think, this explanation does not make the situation clear and a bit more detailed explanation would be desirable.

Author Response

Dear Sir

Thank you very much for your useful comments.

I have revised my manuscript according to your suggestions.

I appreciate if you comfirm my revised manuscript.
